# TOPOLOGY ADAPTIVE GRAPH CONVOLUTIONAL NETWORKS

## ABSTRACT

Convolution acts as a local feature extractor in convolutional neural networks (C-NNs). However, the convolution operation is not applicable when the input data is supported on an irregular graph such as with social networks, citation networks, or knowledge graphs. This paper proposes the topology adaptive graph convolutional network (TAGCN), a novel graph convolutional network that generalizes CNN architectures to graph-structured data and provides a systematic way to design a set of fixed-size learnable filters to perform convolutions on graphs. The topologies of these filters are adaptive to the topology of the graph when they scan the graph to perform convolution, replacing the square filter for the grid-structured data in traditional CNNs. The outputs are the weighted sum of these filters' outputs, extraction of both vertex features and strength of correlation between vertices. It can be used with both directed and undirected graphs. The proposed TAGCN not only inherits the properties of convolutions in CNN for grid-structured data, but it is also consistent with convolution as defined in graph signal processing. Further, as no approximation to the convolution is needed, TAGCN exhibits better performance than existing graph-convolution-approximation methods on a number of data sets. As only the polynomials of degree two of the adjacency matrix are used, TAGCN is also computationally simpler than other recent methods.

## 1 INTRODUCTION

Convolutional neural network (CNN) architectures exhibit state-of-the-art performance on a variety of learning tasks dealing with 1D, 2D, and 3D grid-structured data such as acoustic signals, images, and videos, in which convolution serves as a feature extractor (LeCun et al., 2015). However, the (usual) convolution operation is not applicable when applying CNN to data that is supported on an arbitrary graph rather than on a regular grid structure, since the number of neighbors of each vertex on the graph varies, and it is difficult to design a fixed-size filter scanning over the graph-structured data for feature extraction.

Recently, there has been an increasing interest in graph CNNs (Bruna et al., 2014; Defferrard et al., 2016; Kipf & Welling, 2017; Monti et al., 2017; Levie et al., 2017), attempting to generalize deep learning methods to graph-structured data, specifically focusing on the design of graph CNN . In this paper, we propose the topology adaptive graph convolutional network (TAGCN), a unified convolutional neural network to learn nonlinear representations for the graph-structured data. It slides a set of fixed-size learnable filters on the graph simultaneously, and the output is the weighted sum of these filters' outputs, which extract both vertex features and strength of correlation between vertices. Each filter is adaptive to the topology of the local region on the graph where it is applied. TAGCN unifies filtering in both the spectrum and vertex domains; and applies to both directed and undirected graphs.

In general, the existing graph CNNs can be grouped into two types: spectral domain techniques and vertex domain techniques. In Bruna et al. (2014), CNNs have been generalized to graph-structured data, where convolution is achieved by a pointwise product in the spectrum domain according to the convolution theorem. Later, Defferrard et al. (2016) and Levie et al. (2017) proposed spectrum filtering based methods that utilize Chebyshev polynomials and Cayley polynomials, respectively. The assumption of symmetric adjacency matrix in these spectrum based methods restrict the application to undirected graphs. Kipf & Welling (2017) simplified this spectrum method and obtained

a filter in the vertex domain, which achieves state-of-the-art performance. Other researchers (Atwood & Towsley, 2016; Monti et al., 2017) worked on designing feature propagation models in the vertex domain for graph CNNs. Yang et al. (2016); Dai et al. (2016); Grover & Leskovec (2016); Du et al. (2016) study transforming graph-structured data to embedding vectors for learning problems. Nevertheless, it still remains open how to extend CNNs from grid-structured data to arbitrary graph-structured data with local feature extraction capability.

This paper proposes a modification to the graph convolution step in CNNs that is particularly relevant for graph structured data. Our proposed TAGCN is graph-based convolution and draws on techniques from graph signal processing. We define rigorously the graph convolution operation on the vertex domain as multiplication by polynomials of the graph adjacency matrix, which is consistent with the notion of convolution in graph signal processing. In graph signal processing, polynomials of the adjacency matrix are graph filters, extending to graph based data from the usual concept of filters in traditional time or image based signal processing. Thus, comparing ours with existing work on graph CNNs, our paper provides a solid theoretical foundation for our proposed convolution step instead of an ad-hoc approach to convolution in CNNs for graph structured data.

Further, our method avoids computing the spectrum of the graph Laplacian as in Bruna et al. (2014), or approximating the spectrum using high degree Chebyshev polynomials of the graph Laplacian matrix (in Defferrard et al. (2016), it is suggested that one needs a 25[th] degree Chebyshev polynomial to provide a good approximation to the graph Laplacian spectrum) or using high degree Cayley polynomials of the graph Laplacian matrix (in Levie et al. (2017), 12[th] degree Cayley polynomials are needed). We also clarify that the GCN method in Kipf & Welling (2017) is a first order approximation of the Chebyshev polynomials approximation in Defferrard et al. (2016), which is very different from our method. Our method has a much lower computational complexity than the complexity of the methods proposed in Bruna et al. (2014); Defferrard et al. (2016); Levie et al. (2017), since our method only uses polynomials of the adjacency matrix with maximum degree 2 as shown in our experiments. Finally, the method that we propose exhibits better performance than existing methods. Our contributions are summarized follows:

- We propose a general $K$-localized filter for graph convolution in the vertex domain to extract local features on a set of size-1 up to size-$K$ receptive fields. The topologies of these filters are adaptive to the topology of the graph as they scan the graph to perform convolution. It replaces the fixed square filters in traditional CNNs for the input grid-structured data volumes in traditional CNNs. Thus, our convolution definition that we use in the convolution step for the vertex domain is consistent with convolution in traditional CNNs.

- TAGCN is based on the graph signal processing and it is consistent with the convolution in graph signal processing. It applies to both directed and undirected graphs. Moreover, it has a much lower computational complexity compared with recent methods since it only needs polynomials of the adjacency matrix with maximum degree 2 compared with the 25[th] and 12[th] degree Laplacian matrix polynomials in Defferrard et al. (2016) and Levie et al. (2017).

- As no approximation to the convolution is needed in TAGCN, it achieves better performance compared with existing methods. We contrast TAGCN with recently proposed graph CNN including both spectrum filtering methods (Bruna et al., 2014; Defferrard et al., 2016) and vertex domain propagation methods (Kipf & Welling, 2017; Monti et al., 2017; Atwood & Towsley, 2016), evaluating their performances on three commonly used data sets for graph vertices classification. Our experimental tests show that TAGCN outperforms consistently all other approaches for each of these data sets.

## 2    GRAPH POLYNOMIAL BASED CONVOLUTION ON GRAPHS

We use boldface uppercase and lowercase letters to represent matrices and vectors, respectively. The information and their relationship on a graph $\mathcal{G}$ can be represented by $\mathcal{G} = (\mathcal{V}, \mathcal{E}, \bar{\mathbf{A}})$, where $\mathcal{V}$ is the set of vertices, $\mathcal{E}$ is the set of edges, and $\bar{\mathbf{A}}$ is the weighted adjacency matrix of the graph; the graph can be weighted or unweighted, directed or undirected. We assume there is no isolated vertex in $\mathcal{G}$. If $\mathcal{G}$ is a *directed weighted* graph, the weight $\bar{\mathbf{A}}_{n,m}$ is on the directed edge from vertex $m$ to $n$. The entry $\bar{\mathbf{A}}_{n,m}$ reveals the dependency between node $n$ and $m$ and can take arbitrary real or

complex values. The graph convolution is general and can be adapted to graph CNNs for particular tasks. In this paper, we focus on the vertex semisupervised learning problem, where we have access to limited labeled vertices, and the task is to classify the remaining unlabeled vertices.

## 2.1 GRAPH CONVOLUTION

Without loss of generality, we demonstrate graph convolution on the $\ell$-th hidden layer. The results apply to any other hidden layers. Suppose on the $\ell$-th hidden layer, the input feature map for each vertex of the graph has $C_\ell$ features. We collect the $\ell$-th hidden layer input data on all vertices for the $c$-th feature by the vector $\mathbf{x}_c^{(\ell)} \in \mathbb{R}^{N_\ell}$, where $c = 1, 2, \ldots C_\ell$ and $N_\ell$ is the number of vertices[1]. The components of $\mathbf{x}_c^{(\ell)}$ are indexed by vertices of the data graph representation $\mathcal{G} = (\mathcal{V}, \mathcal{E}, \bar{\mathbf{A}})$[2]. Let $\mathbf{G}_{c,f}^{(\ell)} \in \mathbb{R}^{N_\ell \times N_\ell}$ denote the $f$-th graph filter. The graph convolution is the matrix-vector product, i.e., $\mathbf{G}_{c,f}^{(\ell)} \mathbf{x}_c^{(\ell)}$. Then the $f$-th output feature map is

$$\mathbf{y}_f^{(\ell)} = \sum_{c=1}^{C_\ell} \mathbf{G}_{c,f}^{(\ell)} \mathbf{x}_c^{(\ell)} + b_f \mathbf{1}_{N_\ell}, \tag{1}$$

where $b_f^{(\ell)}$ is a learnable bias, and $\mathbf{1}_{N_\ell}$ is the $N_\ell$ dimension vector of all ones. We design $\mathbf{G}_{c,f}^{(\ell)}$ such that $\mathbf{G}_{c,f}^{(\ell)} \mathbf{x}_c^{(\ell)}$ is a meaningful convolution on a graph with arbitrary topology.

In the recent theory on graph signal processing (Sandryhaila & Moura, 2013), the *graph shift* is defined as a local operation that replaces a graph signal at a graph vertex by a linear weighted combination of the values of the graph signal at the neighboring vertices:

$$\tilde{\mathbf{x}}_c^{(\ell)} = \bar{\mathbf{A}} \mathbf{x}_c^{(\ell)}.$$

The graph shift $\bar{\mathbf{A}}$ extends the time shift in traditional signal processing to graph-structured data. Following Sandryhaila & Moura (2013), a graph filter $\mathbf{G}_{c,f}^{(\ell)}$ is shift-invariant, i.e., the shift $\bar{\mathbf{A}}$ and the filter $\mathbf{G}_{c,f}^{(\ell)}$ commute, $\bar{\mathbf{A}}(\mathbf{G}_{c,f}^{(\ell)} \mathbf{x}_c^{(\ell)}) = \mathbf{G}_{c,f}^{(\ell)}(\bar{\mathbf{A}} \mathbf{x}_c^{(\ell)})$, if under appropriate assumption $\mathbf{G}_{c,f}^{(\ell)}$ is a polynomial in $\mathbf{A}$,

$$\mathbf{G}_{c,f}^{(\ell)} = \sum_{k=0}^{K} g_{c,f,k}^{(\ell)} \mathbf{A}^k. \tag{2}$$

In (2), the $g_{c,f,k}^{(\ell)}$ are the graph filter polynomial coefficients; the quantity $\mathbf{A} = \mathbf{D}^{-\frac{1}{2}} \bar{\mathbf{A}} \mathbf{D}^{-\frac{1}{2}}$ is the normalized adjacency matrix of the graph, and $\mathbf{D} = \text{diag}[\mathbf{d}]$ with the $i$th component being $\mathbf{d}(i) = \sum_j \mathbf{A}_{i,j}$.[3] We adopt the normalized adjacency matrix to guarantee that all the eigenvalues of $\mathbf{A}$ are inside the unit circle, and therefore $\mathbf{G}_{c,f}^{(\ell)}$ is computationally stable. The next subsection shows we will adopt $1 \times C_\ell, 2 \times C_\ell, \ldots$, and $K \times C_\ell$ filters sliding on the graph-structured data. This fact coincides with GoogLeNet (Szegedy et al., 2015), in which a set of filters with different sizes are used in each convolutional layer.

Following the CNN architecture, an additional nonlinear operation, e.g, rectified linear unit (ReLU) is used after every graph convolution operation:

$$\mathbf{x}_f^{(\ell+1)} = \sigma\left(\mathbf{y}_f^{(\ell)}\right),$$

where $\sigma(\cdot)$ denotes the ReLU activation function applied to the vertex values.

In the following subsection, we demonstrate that convolution operator shown in (2) exhibits the nice properties of a square filter in traditional CNNs and generalize the convolution theorem from classical signal processing to signal processing on graphs (Sandryhaila & Moura, 2013).

---

[1]Graph coarsening could be used and the number of vertices may vary for different layers.

[2]We use superscript $(\ell)$ to denote data on the $\ell$th layer and superscript $\ell$ to denote the $\ell$-th power of a matrix.

[3]There is freedom to normalize $\mathbf{A}$ in different ways; here it is assumed that $\bar{\mathbf{A}}_{m,n}$ is nonnegative and the above normalization is well defined.

## 2.2 FILTER DESIGN FOR TAGCN CONVOLUTIONAL LAYER

In this section, we would like to understand the proposed convolution as a feature extraction operator in traditional CNN rather than as propagating labeled data on the graph. Taking this point of view helps us to profit from the design knowledge/experience from traditional CNN and apply it to grid structured data. Our definition of weight of a path and the following filter size for graph convolution in this section make it possible to design a graph CNN architecture similar to GoogLeNet (Szegedy et al., 2015), in which a set of filters with different sizes are used in each convolutional layer. In fact, we found that a combination of size 1 and size 2 filters gives the best performance in all three data sets studied, which is a polynomial with maximum order 2.

In traditional CNN, a $K \times K \times C_\ell$ filter scans over the input grid-structured data for feature extraction. For image classification problems, the value $K$ varies for different CNN architectures and tasks to achieve better performance. For example, in VGG-Verydeep-16 CNN model (Simonyan & Zisserman, 2015), only $3 \times 3 \times C_\ell$ filters are used; in ImageNet CNN model (Krizhevsky et al., 2012), $11 \times 11 \times C_\ell$ filters are adopted; and in GoogLeNet (Szegedy et al., 2015), rather than using the same size filter in each convolutional layer, different size filters, for example, $1 \times 1 \times C_\ell$, $3 \times 3 \times C_\ell$ and $5 \times 5 \times C_\ell$ filters, are concatenated in each convolution layer. Similarly, we propose a general $K$-localized filter for graph CNN.

For a graph-structured data, we cannot use a square filter window since the graph topology is no longer a grid. In the following, we demonstrate that the convolution operation $\mathbf{G}_{c,f}^{(\ell)}\mathbf{x}_c^{(\ell)}$ with $\mathbf{G}_{c,f}^{(\ell)}$ a polynomial filter $\mathbf{G}_{c,f}^{(\ell)} = \sum_{k=0}^{K} g_{c,f,k}^{(\ell)}\mathbf{A}^k$ is equivalent to using a set of filters with filter size from 1 up to $K$. Each $k$-size filter, which is used for local feature extraction on the graph, is $k$-localized in the vertex domain.

Define a *path* of length $m$ on a graph $\mathcal{G}$ as a sequence $v = (v_0, v_1, ..., v_m)$ of vertices $v_k \in \mathcal{V}$ such that each step of the path $(v_k, v_{k+1})$ corresponds to an (directed) edge of the graph, i.e., $(v_k, v_{k+1}) \in \mathcal{E}$. Here one path may visit the same vertex or cross the same edge multiple times. The following adjacency matrix $\mathbf{A}$ is one such example:

$$\mathbf{A} = \begin{bmatrix} 0 & 1 & 0 & 2 & 3 & 0 & 0 & \cdots \\ 1 & 0 & 4 & 5 & 0 & 0 & 0 & \cdots \\ 0 & 1 & 0 & 0 & 0 & 0 & 1 & \cdots \\ 1 & 1 & 0 & 0 & 6 & 0 & 0 & \cdots \\ 1 & 0 & 0 & 1 & 0 & 1 & 0 & \cdots \\ 0 & 0 & 0 & 0 & 1 & 0 & 0 & \cdots \\ 0 & 0 & 0 & 1 & 0 & 0 & 0 & \cdots \\ \vdots & \vdots & \vdots & \vdots & \vdots & \vdots & \vdots & \cdots \end{bmatrix}.$$

Since $\mathbf{A}$ is asymmetric, it represents a directed graph, given in Fig. 2. In this example, there are 6 different length 3-paths on the graph from vertex 2 to vertex 1, namely, $(2, 1, 4, 1)$, $(2, 1, 2, 1)$, $(2, 1, 5, 1)$, $(2, 3, 2, 1)$, $(2, 4, 2, 1)$, and $(2, 4, 5, 1)$.

We further define the *weight of a path* to be the product of the edge weights along the path, i.e., $\phi(p_{0,m}) = \prod_{k=1}^{m} \mathbf{A}_{v_{k-1},v_k}$, where $p_{0,m} = (v_0, v_1, \ldots v_m)$. For example, the weight of the path $(2, 1, 4, 1)$ is $1 \times 1 \times 2 = 2$. Then, the $(i, j)$th entry of $\mathbf{A}^k$ in (2), denoted by $\omega(p_{j,i}^k)$, can be interpreted as the sum of the weights of all the length-$k$ paths from $j$ to $i$, which is

$$\omega(p_{j,i}^k) = \sum_{j \in \{\tilde{j} | \tilde{j} \text{ is } k \text{ paths to } i\}} \phi(p_{j,i}).$$

In the above example, it can be easily verified that $\mathbf{A}_{1,2}^3 = 18$ by summing up the weights of all the above six paths from vertex 2 to vertex 1 with length 3. Then, the $i$th component of $\mathbf{A}^k\mathbf{x}_c^{(\ell)}$ is the weighted sum of the input features of each vertex $\mathbf{x}_c^{(\ell)}$ that are length-$k$ paths away to vertex $i$. Here, $k$ is defined as the *filter size*. The output feature map is a vector with each component given by the size-$k$ filter sliding on the graph following a fixed order of the vertex indices.

The output at the $i$-th component can be written explicitly as $\sum_{c=1}^{C_\ell} \sum_j g_{c,f,k}^{(\ell)} \omega(p_{j,i}^k)\mathbf{x}_c^{(\ell)}(j)$. This weighted sum is similar to the dot product for convolution for a grid-structured data in traditional

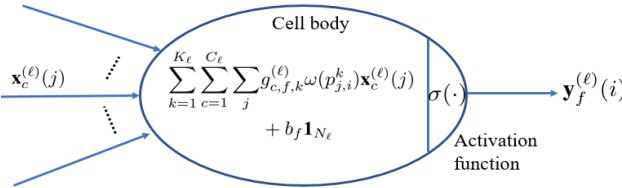

Figure 1: Graph convolution operation at each neuron in the graph convolution layer.

CNNs. Finnaly, the output feature map is a weighted sum of convolution results from filters with different sizes, which is

$$\mathbf{y}_f^{(\ell)}(i) = \sum_{k=1}^{K_\ell} \sum_{c=1}^{C_\ell} \sum_{j \in \{\tilde{j} | \tilde{j} \text{ is } k \text{ paths to } i\}} g_{c,f,k}^{(\ell)} \omega(p_{j,i}^k) \mathbf{x}_c^{(\ell)}(j) + b_f \mathbf{1}_{N_\ell}, \tag{3}$$

The above equation shows that each neuron in the graph convolutional layer is connected only to a local region (local vertices and edges) in the vertex domain of the input data volume, which is adaptive to the graph topology. The strength of correlation is explicitly utilized in $\omega(p_{j,i}^k)$. Fig. 1 summarizes the graph convolution operation at each neuron. We refer to this method as topology adaptive graph convolutional network (TAGCN).

In Fig. 2, we show TAGCN with an example of 2-size filter sliding from vertex 1 (figure on the left-hand-side) to vertex 2 (figure on the right-hand-side). The filter is first placed at vertex 1. Since paths $(1, 2, 1)$ $(5, 4, 1)$ and so on (paths with red glow) are all 2-length paths to vertex 1, they are covered by this 2-size filter. Since paths $(2, 3)$ and $(7, 3)$ are not on any 2-length path to vertex 1, they are not covered by this filter. Further, when this 2-size filter moves to vertex 2, paths $(1, 5)$, $(4, 5)$ and $(6, 5)$ are no longer covered, but paths $(2, 3)$ and $(7, 3)$ are first time covered and contribute to the convolution with output at vertex 2.

Further, $\mathbf{y}_f^{(\ell)}(i)$ is the weighted sum of the input features of vertices in $\mathbf{x}_c^{(\ell)}$ that are within $k$-paths away to vertex $i$, for $k = 0, 1, \dots K$, with weights given by the products of components of $\mathbf{A}^k$ and $g_{c,f,k}^{(\ell)}$. Thus the output is the weighted sum of the feature map given by the filtered results from 1-size up to $K$-size filters. It is evident that the vertex convolution on the graph using $K$th order polynomials is $K$-paths localized. Moreover, different vertices on the graph share $g_{c,f,k}^{(\ell)}$. The above local convolution and weight sharing properties of the convolution (3) on a graph are very similar to those in traditional CNN.

Though the convolution operator defined in (2) is defined on the vertex domain, it can also be understood as a filter in the spectrum domain, and it is consistent with the definition of convolution in graph signal processing. We provide detailed discussion in the appendix.

## 2.3 COMPUTATION COMPLEXITY OF THE CONVOLUTION OPERATION

Equation (7) shows that TAGCN is equivalent to performing graph convolution either on the vertex domain or on the spectrum domain. We next show that it is preferred to implement TAGCN in the vertex domain.

To obtain $\mathbf{F}$, $\mathbf{F}^{-1}$ and $\mathbf{J}$, the eigendecomposition for a diagonalizable matrix is in general $\mathcal{O}(N^3)$. If the graph is directed, $\mathbf{A}$ may only be block diagonalizable, needing the Jordan decomposition, which is sensitive to round-off errors. Additional matrix and vector multiplications may be needed to project the filtered out results from the spectrum domain back to the vertex domain. In contrast, filtering directly in the vertex domain that involves powers of the graph adjacency matrix with $N$ vertices and $M$ edges can be computed in $\mathcal{O}(MN)$ by performing a breadth-first search starting from each vertex to determine the distances to all other vertices Hammack et al. (2011). For a sparse graph, which is often the case in practical problems, $M \ll N$. Since the motivation for performing graph convolution is to detect local features for graph-structured data, the optimal $K$ usually is very small. The experiments section shows that a filter size $K = 2$ leads to the best performance is

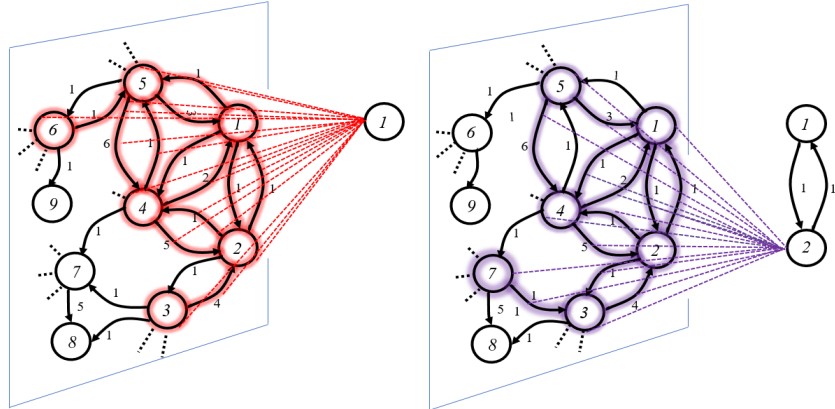

Figure 2: An example of a directed graph with weights along directed edges corresponding to $\mathbf{A}$. The parts with glow on left(right)-hand-side represent filters at different locations. The figure on the left-hand-side denotes the filtering/convolution starting from vertex 1, then the filter slides to vertex 2 as shown on the right-hand-side with filter topology adaptive to the new local region.

selected for the data sets and specific tasks tested. In summary, using polynomial filters in the vertex domain rather than in the spectrum domain saves significantly computational costs.

## 3 RELATION WITH OTHER EXISTING FORMULATIONS

In this section, we show the connections and differences between the proposed TAGCN and existing methods. In general, there are two types of graph convolution operators for the CNN architecture. One defines the convolution in the spectrum domain, whose output feature map is the multiplication of the inverse Fourier transform matrix with the filtered results in the spectrum domain (Bruna et al., 2014; Defferrard et al., 2016; Levie et al., 2017). By doing further approximations based on this spectrum domain operator, a simplified convolution was obtained in Kipf & Welling (2017). The other defines convolution by a feature propagation model in the vertex domain such as MoNet in Monti et al. (2017) and the diffusion CNN (DCNN) in Atwood & Towsley (2016). We investigate in detail each alternative.

In Bruna et al. (2014); Defferrard et al. (2016); Levie et al. (2017), the convolution operation was defined using the convolution theorem and filtering operation in the spectrum domain by computing the eigendecomposition of the normalized Laplacian matrix of the graph. The Laplacian matrix $\mathbf{L}$ is defined as $\mathbf{L} = \mathbf{D} - \mathbf{A}$ with the further assumption that $\mathbf{A}$ is symmetric to guarantee that $\mathbf{L}$ is positive semi-definite. The convolution defined by the multiplication in the spectrum domain is approximated by Defferrard et al. (2016) by

$$\mathbf{U}g_\theta\mathbf{U}^T\mathbf{x} \approx \sum_{k=0}^{K} \theta_k T_k \left[ \frac{2}{\lambda_{\max}}\mathbf{L} - \mathbf{I} \right] \mathbf{x}_c^{(\ell)}, \tag{4}$$

where $T_k[\cdot]$ is the $k$th order matrix Chebyshev polynomial (Shuman et al., 2013) where

$$T_k(\mathbf{L}) = 2\mathbf{L}T_{k-1}[\mathbf{L}] - T_{k-2}[\mathbf{L}], \tag{5}$$

with the initial values defined as $T_0[\mathbf{L}] = \mathbf{I}$ and $T_1[\mathbf{L}] = \mathbf{L}$. We refer later to this method as ChebNet for performance comparison. Note that the Laplacian matrix can be seen as a differentiator operator. The assumption of symmetric $\mathbf{A}$ restricts the application to undirected graphs. Note that in Defferrard et al. (2016), Laplacian matrix polynomials with maximum order $K = 25$ is needed to approximate the convolution operation on the left-hand side in (4), which imposes the computational burden. While TAGCN only needs an adjacency matrix polynomials with maximum order 2 to achieve better performance as shown in the experiment part.

In Kipf & Welling (2017), a graph convolutional network (GCN) was obtained by a first order approximation of (4). In particular, let $K = 1$ and make the further assumptions that $\lambda_{\max} = 2$ and

$\theta_0 = \theta_1 = \theta$. Then a simpler convolution operator that does not depend on the spectrum knowledge is obtained as

$$\mathbf{U}g_\theta\mathbf{U}^T\mathbf{x} \approx \sum_{k=0}^{1}\theta_k T_k\left[\frac{2}{\lambda_{\max}}\mathbf{L}-\mathbf{I}\right]\mathbf{x}_c^{(\ell)} \approx \theta(\mathbf{I}+\mathbf{D}^{-\frac{1}{2}}\mathbf{A}\mathbf{D}^{-\frac{1}{2}})\mathbf{x}_c^{(\ell)}.$$

Note that $\mathbf{I}+\mathbf{D}^{-\frac{1}{2}}\mathbf{A}\mathbf{D}^{-\frac{1}{2}}$ is a matrix with eigenvalues in $[0,2]$. A renormalization trick is adopted here by letting $\widetilde{\mathbf{A}} = \mathbf{A}+\mathbf{I}$ and $\widetilde{\mathbf{D}}_{i,i} = \sum_j \widetilde{\mathbf{A}}_{i,j}$. Finally, the convolutional operator is approximated by

$$\mathbf{U}g_\theta\mathbf{U}^T\mathbf{x} \approx \theta\widetilde{\mathbf{D}}^{-\frac{1}{2}}\widetilde{\mathbf{A}}\widetilde{\mathbf{D}}^{-\frac{1}{2}}\mathbf{x}_c^{(\ell)} = \theta\widehat{\mathbf{A}}, \tag{6}$$

where $\widehat{\mathbf{A}} = \widetilde{\mathbf{D}}^{-\frac{1}{2}}\widetilde{\mathbf{A}}\widetilde{\mathbf{D}}^{-\frac{1}{2}}$. It is interesting to observe that this method though obtained by simplifying the spectrum method has a better performance than the spectrum method (Defferrard et al., 2016). The reason may be because the simplified form is equivalent to propagating vertex features on the graph, which can be seen as a special case of our TAGCN method, though there are other important differences.

Though our TAGCN is able to leverage information at a farther distance, it is not a simple extension of GCN Kipf & Welling (2017). First, the graph convolution in GCN is defined as a first order Chebyshev polynomial of the graph Laplacian matrix, which is an approximation to the graph convolution defined in the spectrum domain in Defferrard et al. (2016). In contrast, our graph convolution is rigorously defined as multiplication by polynomials of the graph adjacency matrix; this is not an approximation, rather, it simply is filtering with graph filters as defined and as being consistent with graph signal processing.

Next, we show the difference between our work and the GCN method in Kipf & Welling (2017) when using 2nd order (K=2, 2 steps away from the central node) Chebyshev polynomials of Laplacian matrix. In the GCN paper Kipf & Welling (2017), it has been shown that $\sum_{k=0}^{1}\theta_k T_k(\mathbf{L}) \approx \widehat{\mathbf{A}}$ as repeated in (6), and $T_2[\mathbf{L}] = 2\mathbf{L}^2$ by the definition of Chebyshev polynomial. Then, extending GCN to the second order Chebyshev polynomials (two steps away from a central node) can be obtained from the original definition in T. Kipfs GCN (Kipf & Welling, 2017, eqn (5)) as $\sum_{k=0}^{2}\theta T_k(L) = \widehat{\mathbf{A}} + 2\mathbf{L}^2 - \mathbf{I}$, which is different from our definition as in (2). Thus, it is evident that our method is not a simple extension of GCN. We apply graph convolution as proposed from basic principles in the graph signal processing, with no approximations involved, while both GCN in Kipf & Welling (2017) and Defferrard et al. (2016) Levie et al. (2017) are based on approximating the convolution defined in the spectrum domain. In our approach, the degree of freedom is the design of the graph filter-its degree and its coefficients. Ours is a principled approach and provides a generic methodology. The performance gains we obtain are the result of capturing the underlying graph structure with no approximation in the convolution operation.

Simonovsky & Komodakis (2017) proposed the edge convolution network (ECC) to extend the convolution operator from regular grids to arbitrary graphs. The convolution operator is defined similarly to (6) as

$$\mathbf{y}_f^{(\ell)}(i) = \sum_{j\in\mathcal{N}(i)}\frac{1}{|\mathcal{N}(i)|}\mathbf{\Theta}_{j,i}^{(\ell)}\mathbf{x}_c^{(\ell)}(j) + b_f^{(\ell)},$$

with $\mathbf{\Theta}_{j,i}^{(\ell)}$ is the weight matrix that needs to be learned.

A mixture model network (MoNet) was proposed in Monti et al. (2017), with convolution defined as

$$\mathbf{y}_f^{(\ell)}(i) = \sum_{f=1}^{F}\sum_{j\in\mathcal{N}(i)}g_f\kappa_f\mathbf{x}_c^{(\ell)}(j),$$

where $\kappa_f$ is a Gaussian kernel with $\kappa_f = \exp\left\{-\frac{1}{2}(\mathbf{u}-\boldsymbol{\mu}_f)^T\mathbf{\Sigma}_f^{-1}(\mathbf{u}-\boldsymbol{\mu}_f)\right\}$ and $g_f$ is the weight coefficient for each Gaussian kernel $\kappa_f$. It is further assumed that $\mathbf{\Sigma}_f$ is a $2\times 2$ diagonal matrix.

GCN, ECC, and MoNet all design a propagation model on the graph; their differences are on the weightings used by each model.

Table 1: Summary of number of weights need to be learned for the $\ell$-th layer.

| DCNN | ECC | ChebNet | GCN | MoNet | TAGCN |
|------|-----|---------|-----|-------|-------|
| $F_\ell C_\ell$ | $F_\ell C_\ell$ | $25F_\ell C_\ell$ | $F_\ell C_\ell$ | $4F_\ell C_\ell$ | $2F_\ell C_\ell$ |

Atwood & Towsley (2016) proposed a diffusion CNN (DCNN) method that considers a diffusion process over the graph. The transition probability of a random walk on a graph is given by $\mathbf{P} = \mathbf{D}^{-1}\mathbf{A}$, which is equivalent to the normalized adjacency matrix.

$$\mathbf{y}_{c,f}^{(\ell)} = \mathbf{g}_{c,f}^{(\ell)}\mathbf{P}^f\mathbf{x}_c^{(\ell)}.$$

By comparing the above methods with TAGCN in (3), it can be concluded that GCN, ECC and MoNet can be seen as a special case of TAGCN because in (3) the item with $k = 1$ can be seen as an information propagation term. However, the strength of the correlation between vertices that are taken into account in $\omega(p_{j,i}^k)$ in (3) can not be utilized in the GCN, ECC and MoNet methods. Further, the TAGCN method is a systematic way to design a set of fixed size filters that is adaptive to the input graph topology when performing convolution on the graph. Compared with existing spectrum methods (Bruna et al., 2014; Defferrard et al., 2016; Levie et al., 2017), TAGCN satisfies the convolution theorem as shown in the previous subsection and implemented in the vertex domain, which avoids performing costly and practically numerical unstable eigendecompositions.

We further compare the number of weights that need to be learned in each hidden layer for these different methods in Table 1. As we show later, $K = 2$ is selected in our experiments using cross validation. However, for ChebNet in (Defferrard et al., 2016), it is suggested that one needs a $25^{\text{th}}$ degree Chebyshev polynomial to provide a good approximation to the graph Laplacian spectrum. Thus we have a moderate number of weights to be learned. In the following, we show that our method achieves the best performance for each of those commonly used graph-structured data sets.

## 4 EXPERIMENTS

The proposed TAGCN is general and can be fit to the general graph CNN architectures for different tasks. In the experiments, we focus on the vertex semisupervised learning problem, where we have access to only a few labeled vertices, and the task is to classify the remaining unlabeled vertices. To compare the performance of TAGCN with that of existing methods, we extensively evaluate TAGCN on three graph-structured datasets, including the Cora, Citesser and Pubmed datasets. The datasets split and experiments setting closely follow the standard criteria in Yang et al. (2016).

### 4.1 DATASETS

TAGCN is evaluated on three data sets coming from citation networks: Citeseer, Cora, and Pubmed. We closely follow the data set split and experimental setup in Yang et al. (2016). Each data set consists of a certain classes of documents, yet only a few documents are labeled. The task is to classify the documents in the test set with these limited number of labels. In each data set, the vertices are the documents and the edges are the citation links. Each document is represented by sparse bag-of-words feature vectors, and the citation links between documents are provided. Detailed statistics of these three data sets are summarized in Table 2. It shows the number of nodes and edges that corresponding to documents and citation links. Classes for the graph vertices shows the number of document classes in each data set. Also, the number of features at each vertex is given. Label rate denotes the number of labeled documents that are used for training divided by the total number of documents in each data set.

Table 2: Dataset statistics, following Yang et al. (2016); Kipf & Welling (2017)

| Dataset Size | Nodes | Edges | Classes | Features | Label rate |
|---|---|---|---|---|---|
| Citeseer | 3,327 | 4,732 | 6 | 3,703 | 0.036 |
| Cora | 2,708 | 5,429 | 7 | 1,433 | 0.052 |
| Pubmed | 19,717 | 44,338 | 3 | 500 | 0.003 |

Table 3: Summary of results in terms of percentage classification accuracy with standard variance

| | Citeseer | Cora | Pubmed |
|---|---|---|---|
| Planetoid | 64.7 | 75.7 | 77.2 |
| DCNN | - | 76.8±0.6 | 73.0±0.5 |
| ChebNet (K=2) | 69.6 | 81.2 | 73.8 |
| ChebNet (K=3) | 69.8 | 79.5 | 74.4 |
| GCN | 70.3 | 81.5 | 79.0 |
| MoNet | - | 81.69±0.5 | 78.81±0.4 |
| TAGCN (K=2) | **70.9± 0.9** | **82.5±0.7** | **81.1±0.4** |

## 4.2 EXPERIMENTAL SETTINGS

We construct a graph for each data set with nodes representing documents and undirected edges[4] linking two papers if there is a citation relationship. We obtain the adjacency matrix $\bar{\mathbf{A}}$ with $0$ and $1$ components and further obtain the normalized matrix $\mathbf{A}$.

In the following experiments, we design a TAGCN with two hidden layers (obtained from cross validation) for the semi-supervised node classification. In each hidden layer, the proposed TAGCN is applied for convolution, followed by a ReLU activation. 16 hidden units (filters) are designed for each hidden layer, and dropout is applied after each hidden layer. The softmax activation function is applied to the output of the second hidden layer for the final classification. For ablation study, we evaluate the performance of TAGCN with different kernel size from 1 to 4. To investigate the performance for different number of parameters, we also design a TAGCN with $8$ filters for each hidden layer and compare its classification accuracy with all the baselines and TAGCN with $16$ filters. We train our model using Adam (Kinga & Ba, 2015) with a learning rate of $0.01$ and early stopping with a window size of $20$. Hyperparameters of the networks (filter size, dropout rate, and number of hidden layers) are selected by cross validation.

To make a fair comparison, we closely follow the same split of training, validation, and testing sets as in Kipf & Welling (2017); Yang et al. (2016), i.e., $500$ labeled examples for hyperparameters (filter size, dropout rate, and number of hidden layers) optimization and cross-entropy error is used for classification accuracy evaluation. The performance results of the proposed TAGCN method are an average over 100 runs with random initializations (Glorot & Bengio, 2010).

## 4.3 QUANTITATIVE EVALUATIONS

We compare the classification accuracy with other recently proposed graph CNN methods as well as a graph embedding method known as Planetoid Yang et al. (2016). The quantitative results are summarized in Table 3. Reported numbers denote classification accuracy in percent. Results for Planetoid, GCN, and ChebNet are taken from Kipf & Welling (2017), and results for DCNN and MoNet are taken from Monti et al. (2017). All the experiments for different methods are based on the same data statistics shown in Table 2. The datasets split and experiments settings closely follow the standard criteria in Yang et al. (2016); Kipf & Welling (2017). Table 3 shows that our method

---

[4]We use undirected graphs here as citation relationship gives positive correlation between two documents. However, in contrast with the other approaches surveyed here, the TAGCN method is not limited to undirected graphs if directed graphs are better suited to the applications.

Table 4: TAGCN classification accuracy (ACC) with different parameters

| Data Set | Filter Size | Filter Number | ACC |
|---|---|---|---|
| Citeseer | 1 | 16 | 68.9 |
| | 2 | 16 | **70.9** |
| | 3 | 16 | 70.0 |
| | 4 | 16 | 69.8 |
| | 2 | **8** | **70.6** |
| Cora | 1 | 16 | 81.4 |
| | 2 | 16 | **82.5** |
| | 3 | 16 | 82.1 |
| | 4 | 16 | 81.8 |
| | 2 | **8** | **82.5** |
| Pubmed | 1 | 16 | 79.4 |
| | 2 | 16 | **81.1** |
| | 3 | 16 | 80.9 |
| | 4 | 16 | 79.5 |
| | 2 | **8** | **80.8** |

outperforms all the recent state-of-the-art methods by obvious margins for all the three datasets. These results verify the efficacy of the proposed TAGCN.

For ablation study, we further compare the performance of different filter sizes from $K = 1$ to $K = 4$ in Table 4. It shows that the performances for filter size $K = 2$ are always better than that for other filter sizes. The value $K = 1$ gives the worst classification accuracy. This further validates that a local feature extraction is more important than just propagating features among neighbors on the graph. In Table 4, we also compare the performance of different number of filters, which reflects different number of network parameters. Note, we also choose filer size $K = 2$ and filter number $F_\ell = 8$ that results in the same number of network parameters as that in GCN, MoNet, ECC and DCNN according to Table 1. It shows that the classification accuracy using 8 filters is comparable with that using 16 filters in each hidden layer for TAGCN. Moreover, TAGCN with 8 filters can still achieve higher accuracy than GCN, MoNet, ECC and DCNN methods. This proves that, even with a similar number of parameters or architecture, our method still exhibits superior performance than GCN.

As we explained in Section 3, TAGCN in our paper is not simply extending GCN Kipf & Welling (2017) to $k$-th order. Nevertheless, we implement $\mathbf{A}^2$ and compare its performance with ours. For the data sets Pubmed, Cora, and Citeseer, the classification accuracies are 79.1(81.1), 81.7(82.5) and 70.8(70.9), where the numbers in parentheses are the results obtained with our method. Our method still achieves a noticeable performance advantage over $\mathbf{A}^2$ for the Pubmed and Cora data; in particular, we note the significant performance gain with the Pubmed database that has the largest number of nodes among these three data sets.

## 5 CONCLUSIONS

We have defined a novel graph convolutional network that rearchitects the CNN architecture for graph-structured data. The proposed method, known as TAGCN, is adaptive to the graph topology as the filter scans the graph. Further, TAGCN inherits properties of the convolutional layer in classical CNN, i.e., local feature extraction and weight sharing. It can further extract the strength of correlation between vertices in the filtering region. On the other hand, by the convolution theorem, TAGCN that implements in the vertex domain offers implement in the spectrum domain unifying graph CNN in both the spectrum domain and the vertex domain. TAGCN is consistent with convolution in graph signal processing. These nice properties lead to a noticeable performance advantage in classification accuracy on different graph-structured datasets for semi-supervised graph vertex classification problems with low computational complexity.

## 6    APPENDIX: SPECTRUM RESPONSE OF TAGCN

In classical signal processing (Oppenheim & Schafer, 2009), the convolution in the time domain is equivalent to multiplication in the spectrum domain. This relationship is known as the convolution theorem. Sandryhaila & Moura (2013) showed that the graph filtering defined in the vertex domain satisfies the generalized convolution theorem naturally and can also interpret spectrum filtering for both directed and undirected graphs. Recent work (Bruna et al., 2014; Defferrard et al., 2016) used the convolution theorem for undirected graph-structured data and designed a spectrum graph filtering.

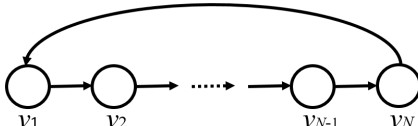

Figure 3: Graph topology of a 1-D cyclic graph.

Assume that the adjacency matrix $\mathbf{A}$ for a graph is diagonalizable, i.e., $\mathbf{A} = \mathbf{F}^{-1}\mathbf{J}\mathbf{F}$ with $\mathbf{J}$ a diagonal matrix. The components on the diagonal of $\mathbf{J}$ are eigenvalues of $\mathbf{A}$, and the column vectors of $\mathbf{F}^{-1}$ are the right eigenvectors of $\mathbf{A}$; the row vectors of $\mathbf{F}$ are the left eigenvectors of $\mathbf{A}$ [5]. By diagonalizing $\mathbf{A}$ in (2) for TAGCN, we obtain

$$\mathbf{G}_{c,f}^{(\ell)}\mathbf{x}_c^{(\ell)} = \mathbf{F}^{-1}\left(\sum_{k=0}^{K} g_{c,f,k}^{(\ell)}\mathbf{J}^k\right)\mathbf{F}\mathbf{x}_c^{(\ell)}. \tag{7}$$

The expression on the left-hand-side of the above equation represents the filtering/convolution on the vertex domain. Matrix $\mathbf{F}$ defines the graph Fourier transform (Sandryhaila & Moura, 2013; 2014), and $\mathbf{F}\mathbf{x}_c^{(\ell)}$ is the input feature spectrum map, which is a linear mapping from the input feature on the vertex domain to the spectrum domain. The polynomial $\sum_{k=0}^{K} g_{c,f,k}^{(\ell)}\mathbf{J}^k$ is the spectrum of the graph filter. Relation (7), which is equation (27) in Sandryhaila & Moura (2013) generalizes the classical convolution theorem to graph-structured data: convolution/filtering on the vertex domain becomes multiplication in the spectrum domain. When the graph is in the 1D cyclic form, as shown in Fig. 3, the corresponding adjacency matrix is of the form

$$\mathbf{A} = \begin{bmatrix} & & & 1 \\ 1 & & & \\ & \ddots & & \\ & & 1 & \end{bmatrix}.$$

The eigendecomposition of $\mathbf{A}$ is

$$\mathbf{A} = \frac{1}{N}\text{DFT}^{-1}\begin{bmatrix} e^{-j\frac{2\pi 0}{N}} & & \\ & \ddots & \\ & & e^{-j\frac{2\pi(N-1)}{N}} \end{bmatrix}\text{DFT},$$

where DFT is the discrete Fourier transform matrix. The convolution operator defined in (2) is consistent with that in classical signal processing.

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
