# OpenReview forum: "Topology Adaptive Graph Convolutional  Networks"
_ICLR.cc/2018/Conference — Reject_

### Official Review · AnonReviewer2 · 2017-11-27
**Major revision is recommended**

**Rating:** 4
**Confidence:** 4

**Review:**

In this paper a new neural network architecture for semi-supervised graph classification is proposed. The new construction builds upon graph polynomial filters and utilizes them on each successive layer of the neural network with ReLU activation functions.

In my opinion writing of this paper requires major revision. The first 8 pages mostly constitute a literature review and experimental section provides no insights about the performance of the TAGCN besides the slight improvement of the Cora, Pubmed and Citeseer benchmarks.

The one layer analysis in sections 2.1, 2.2 and 2.3 is simply an explanation of graph polynomial filters, which were previously proposed and analyzed in cited work of Sandryhaila and Moura (2013). Together with the summary of other methods and introduction, it composes the first 8 pages of the paper. I think that the graph polynomial filters can be summarized in much more succinct way and details deferred to the appendix for interested reader. I also recommend stating which ideas came from the Sandryhaila and Moura (2013) work in a more pronounced manner.

Next, I disagree with the statement that "it is not clear how to keep the vertex local property when filtering in the spectrum domain". Graph Laplacian preserves the information about connectivity of the vertices and filtering in the vertex domain can be done via polynomial filters in the Fourier domain. See Eq. 18 and 19 in [1].

Finally, I should say that TAGCN idea is interesting. I think it can be viewed as an extension of the GCN (Kipf and Welling, 2017), where instead of an adjacency matrix with self connections (i.e. first degree polynomial), a higher degree graph polynomial filter is used on every layer (please correct me if this comparison is not accurate). With more experiments and interpretation of the model, including some sort of multilayer analysis, this can be a good acceptance candidate.


[1] David I Shuman, Sunil K Narang, Pascal Frossard, Antonio Ortega, and Pierre Vandergheynst.
The emerging field of signal processing on graphs: Extending high-dimensional data analysis to
networks and other irregular domains. IEEE Signal Processing Magazine, 30(3):83–98, 2013.

---

> ### Author Response · Authors · 2017-12-30
> **Reply to AnonReviewer 2: "Major revision is recommended" (Q1-Q3/Q4)**
>
> 1)	"In my opinion writing of this paper requires major revision. The first 8 pages mostly constitute a literature review and experimental section provides no insights about the performance of the TAGCN besides the slight improvement of the Cora, Pubmed and Citeseer benchmarks. "
>
> Reply: We have reorganized the paper and added more insights for the proposed TAGCN algorithm. We explain our proposed method in Section 2 and compare it with previous work in Section 3 to emphasize the novelty and differences of our method. We want to emphasize that the adjacency matrix polynomial filter (graph convolution operation) defined on the vertex domain (our method) is totally different from all the existing graph CNN methods available and that define the convolution in the spectrum domain. Thus, our proposed convolution, its computational complexity, and understanding of the choice of the filter size, all need adequate explanations, and these are given in Section 3. Even some of the reviewers seem to misunderstand the GCN method in Kipf & Welling 2017 based on approximations by matrix Chebyshev polynomials in Defferrard et al. 2016 with our method. Thus explaining adequately our method from different perspectives is necessary. We have further described these relationships in Section 3 in the revised version and made the architecture of our method clearer.
>
> As for performance, our method outperforms all recently proposed methods on all three datasets. In addition, for the Pubmed dataset, which is much larger than the Citeseer and Cora data sets, we have a 2.1% improvement over GCN (Kipf & Welling 2017) and 6.7% improvement over ChebNet (Defferrard et al. 2016).  These performance results are averages obtained over 100 Monte Carlo runs.  As far as we know and as far as we can determine, our method exhibits the best performance on the Pubmed data not only when compared with all previous available publications, as well as when compared with all papers submitted to ICLR18, see papers below. Also, please note that, as explained by the authors, the last paper listed below fails with the Pubmed data set because of its storage complexity.
>
> Graph Partition Neural Networks for Semi-Supervised Classification, submitted to ICLR18
> Attention-based Graph Neural Network for Semi-supervised Learning, submitted to ICLR18
> Stochastic Training of Graph Convolutional Networks, submitted to ICLR18
> Graph Attention Networks, submitted to ICLR18
>
> 2)  "The one layer analysis in sections 2.1, 2.2 and 2.3 is simply an explanation of graph polynomial filters, which were previously proposed and analyzed in cited work of Sandryhaila and Moura (2013). Together with the summary of other methods and introduction, it composes the first 8 pages of the paper. I think that the graph polynomial filters can be summarized in much more succinct way and details deferred to the appendix for interested reader. I also recommend stating which ideas came from the Sandryhaila and Moura (2013) work in a more pronounced manner. "
>
> Reply: Thank you for your suggestion. We have moved Section 2.3 to the Appendix following your suggestion. Sections 2.1 and 2.2 explain important concepts in our proposed method: the definition of graph CNN, graph filter size, as well as how to understand graph convolution as a local feature extractor, which are important for the understanding of our graph CNN and do not appear in Sandryhaila and Moura (2013). We better describe these subsections in the revised version and make them more succinct and clearer.
>
> 3) "Next, I disagree with the statement that "it is not clear how to keep the vertex local property when filtering in the spectrum domain". Graph Laplacian preserves the information about connectivity of the vertices and filtering in the vertex domain can be done via polynomial filters in the Fourier domain. See Eq. 18 and 19 in [1]. "
>
> [1] David I Shuman, Sunil K Narang, Pascal Frossard, Antonio Ortega, and Pierre Vandergheynst. The emerging field of signal processing on graphs: Extending high-dimensional data analysis to networks and other irregular domains. IEEE Signal Processing Magazine, 30(3):83–98, 2013.
>
> Reply: Thank you for pointing out this. We have removed this sentence and referred to [1] in the revised version.

---

> > ### Author Response · Authors · 2017-12-30
> > **Reply to AnonReviewer 2: "Major revision is recommended" (Q4-Q4/Q4)**
> >
> > 4) "Finally, I should say that TAGCN idea is interesting. I think it can be viewed as an extension of the GCN (Kipf and Welling, 2017), where instead of an adjacency matrix with self connections (i.e. first degree polynomial), a higher degree graph polynomial filter is used on every layer (please correct me if this comparison is not accurate). With more experiments and interpretation of the model, including some sort of multilayer analysis, this can be a good acceptance candidate. "
> >
> > Reply: Thank you for your encouraging comments. We have added more interpretation of the model in the revised version.  We have also done further experiments for the filter of A^2. As we explained below in the next paragraph, graph convolution in our paper is not simply extending GCN to k-th order. Nevertheless, we implemented A^2 and compared its performance with ours. For the data sets Pubmed, Cora, and Citeseer, the classification accuracies are 79.1 (81.1), 81.7(82.5) and 70.8 (70.9), where the numbers in parentheses are the results obtained with our method. Our method still achieves a noticeable performance advantage over A^2 for the Pubmed and Cora data; in particular, we note the significant performance gain with the Pubmed database that has the largest number of nodes among these three data sets. For multi-layer analysis, we did experiment by further extending the hidden layers. However, there is no performance improvement, which is consistent with the multilayer analysis in Kipf & Welling 2017.
> >
> > We agree with the reviewer that our method is able to leverage information at a farther distance on the graph than the GCN (Kipf & Welling 2017). However, ours is not a simple extension of GCN. In fact, extending GCN to second order would not lead to our results. We clarify the fundamental difference between our method and the GCN methodology if we extended the latter to a higher order in a separate comment (due to space limitation) with title “Differences between the proposed TAGCN and GCN in Kipf & Welling 2017”. Thank you for your attention. We have added the corresponding discussion in Section 3 of the revised version.

---

### Official Review · AnonReviewer3 · 2017-11-27
**Reasonable technical contribution, solid evaluation, well written**

**Rating:** 6
**Confidence:** 3

**Review:**

The authors propose a new CNN approach to graph classification that generalizes previous work. Instead of considering the direct neighborhood of a vertex in the convolution step, a filter based on outgoing walks of increasing length is proposed. This incorporates information from more distant vertices in one propagation step.

The proposed idea is not exceptional original, but the paper has several strong points:

* The relation to previous work is made explicit and it is show that several previous approaches are generalized by the proposed one.
* The paper is clearly written and well illustrated by figures and examples. The paper is easy to follow although it is on an adequate technical level.
* The relation between the vertex and spectrum domain is well elaborated and nice (although neither important for understanding nor implementing the approach).
* The experimental evaluation appears to be sound. A moderate improvement compared to other approaches is observed for all data sets.

In summary, I think the paper can be accepted for ICLR.
----------- EDIT -----------
After reading the publications mentioned by the other reviewers as well as the following related contributions

* Network of Graph Convolutional Networks Trained on Random Walks (under review for ICLR 2018)
* Graph Convolution: A High-Order and Adaptive Approach, Zhenpeng Zhou, Xiaocheng Li (arXiv:1706.09916)

I agree that the relation to previous work is not adequately outlined. Therefore I have modified my rating accordingly.

---

> ### Author Response · Authors · 2017-12-30
> **Reply to AnonReviewer 3: "Reasonable technical contribution, solid evaluation, well written"**
>
> Thank you for the positive comments.
>
> 1) We agree with the reviewer that our method is able to leverage information at a farther distance on the graph than the GCN of Kipf & Welling 2017. However, ours is not a simple generalization of GCN. In fact, extending GCN to the second order would not lead to our results. We clarify the fundamental difference between our method and the GCN methodology if we extended the latter to a higher order in the separate comment with title “Differences between the proposed TAGCN and GCN in Kipf & Welling 2017” due to space limitation. Thank you for your attention. We have added the corresponding discussion in Section 3 of the revised version.
>
> 2) "After reading the publications mentioned by the other reviewers as well as the following related contributions
> * Network of Graph Convolutional Networks Trained on Random Walks (under review for ICLR 2018)
> * Graph Convolution: A High-Order and Adaptive Approach, Zhenpeng Zhou, Xiaocheng Li (arXiv:1706.09916)
> I agree that the relation to previous work is not adequately outlined. Therefore I have modified my rating accordingly."
>
> We thank the reviewer for pointing out these two recent works on graph CNN. We would like to point out that our method is substantially different from these two papers.
>
> The graph convolutions in these two papers are defined based on ad-hoc methods, which do not have the physical meaning of convolution. The first paper concatenates A^k with k from 0 to 6, and the second paper defines {\tilde A}^k = min{A^k + I,1}. In contrast, our definition of convolution is based on graph signal processing, it is consistent with the convolutional theorem, and, finally, it reduces to classical convolution for the direct circle topology.
>
> The first paper reports their top 3 performers rather than reporting average performance, while we report an averaged performance over 100 trails. Our performance for Pubmed is still better than theirs even if the comparison is unfair. The second paper did not follow the usual data splitting method and so we cannot compare ours to their performance directly.
>
> We respectfully disagree with the comment that these two papers are not adequately outlined in the original submission. The first paper is submitted to the same conference ICLR2018 as our paper, so, at the same time – how could we have access to it before hand? Thus, there is no way we could refer to this paper in our original submission. The second paper appeared in arXiv with the title “Graph Convolutional Networks for Molecules,” which was specific to molecules with content that was quite different from its second version. The second version was submitted to arXiv on Oct, 20, becoming only available to the public almost at the same time as we submitted our paper to ICLR2018. Further, there is a major revision between these two versions as we can see on arXiv, and the number of pages increased from less than 5 pages and a half to 8 pages. We thank the reviewer for finding this paper for us.
>
> We also want to mention that, besides providing a solid foundation for our proposed the graph convolution operation, our method also exhibits better performance due to the fact that no approximation is needed for the convolution operation. Our method outperforms all recently proposed methods on all three datasets. In addition, for the Pubmed dataset, which is much larger than the Citeseer and Cora data sets, we have a 2.1% improvement over GCN (Kipf & Welling 2017) and 6.7% improvement over ChebNet (Defferrard et al. 2016).  These performance results are averages obtained over 100 Monte Carlo runs.  As far as we know and as far as we can determine, our method exhibits the best performance on the Pubmed data not only when compared with all previous available publications, as well as when compared with all papers submitted to ICLR18, see papers below. Also, please note that, as explained by the authors, the last paper listed below fails with the Pubmed data set because of its storage complexity.
>
> Graph Partition Neural Networks for Semi-Supervised Classification, submitted to ICLR18
> Attention-based Graph Neural Network for Semi-supervised Learning, submitted to ICLR18
> Stochastic Training of Graph Convolutional Networks, submitted to ICLR18
> Graph Attention Networks, submitted to ICLR18

---

> > ### Comment · AnonReviewer3 · 2018-01-12
> > **Initial submission has been improved**
> >
> > Thank you for pointing out that the Arxiv paper was updated recently. Please note that I did not mean to require that the two mentioned articles should have been referenced and discussed in your initial submission.
> >
> > I understand that there are technical differences between TAGCN and higher-order extensions of GCN, but still both approaches pursue the goal to incorporate information from nodes at a farther distance from the reference node and are very similar in that sense. Thank you for adding additional experimental results on this. I have changed my rating back to its original value.

---

### Official Review · AnonReviewer1 · 2017-11-27
**Interesting improvement idea, clarity could be improved**

**Rating:** 5
**Confidence:** 4

**Review:**

The paper introduces Topology Adaptive GCN (TAGCN) to generalize convolutional
networks to graph-structured data.
I find the paper interesting but not very clearly written in some sections,
for instance I would better explain what is the main contribution and devote
some more text to the motivation. Why is the proposed approach better than the
previously published ones, and when is that there is an advantage in using it?

The main contribution seems to be the use of the "graph shift" operator from
Sandryhaila and Moura (2013), which closely resembles the one from
Shuman et al. (2013). It is actually not very well explained what is the main
difference.

Equation (2) shows that the learnable filters g are operating on the k-th power
of the normalized adjacency matrix A, so when K=1 this equals classical GCN
from T. Kipf et al.
By using K > 1 the method is able to leverage information at a farther distance
from the reference node.

Section 2.2 requires some polishing as I found hard to follow the main story
the authors wanted to tell. The definition of the weight of a path seems
disconnected from the main text, ins't A^k kind of a a diffusion operator or
random walk?
This makes me wonder what would be the performance of GCN when the k-th power
of the adjacency is used.

I liked Section 3, however while it is true that all methods differ in the way they
do the filtering, they also differ in the way the input graph is represented
(use of the adjacency or not).

Experiments are performed on the usual reference benchmarks for the task and show
sensible improvements with respect to the state-of-the-art. TAGCN with K=2 has
twice the number of parameters of GCN, which makes the comparison not entirely
fair. Did the author experiment with a comparable architecture?
Also, how about using A^2 in GCN or making two GCN and concatenate them in
feature space to make the representational power comparable?

It is also known that these benchmarks, while being widely used, are small and
result in high variance results. The authors should report statistics over
multiple runs.
Given the systematic parameter search, with reference to the actual validation
(or test?) set I am afraid there could be some overfitting. It is quite easy
to probe the test set to get best performance on these benchmarks.

As a minor remark, please make figures readable also in BW.

Overall I found the paper interesting but also not very clear at pointing out
the major contribution and the motivation behind it. At risk of being too reductionist:
it looks as learning a set of filters on different coordinate systems given
by the various powers of A. GCN looks at the nearest neighbors and the paper
shows that using also the 2-ring improves performance.

---

> ### Author Response · Authors · 2017-12-30
> **Reply to AnonReviewer 1: "Interesting improvement idea, clarity could be improved"-(Q1/Q11)**
>
> 1) "I find the paper interesting but not very clearly written in some sections, for instance I would better explain what is the main contribution and devote some more text to the motivation. Why is the proposed approach better than the previously published ones, and when is that there is an advantage in using it?"
>
> Reply: Thank you for your suggestion. We now provide additional explanation of the main contributions and strengths of our approach in the revised version. This paper proposes a modification to the graph convolution step in CNNs that is particularly relevant for graph structured data. Our proposed convolution is graph-based convolution and draws on techniques from graph signal processing. We define rigorously the graph convolution operation on the vertex domain as multiplication by polynomials of the graph adjacency matrix, which is consistent with the notion of convolution in graph signal processing. In graph signal processing, polynomials of the adjacency matrix are graph filters, extending to graph based data the usual concept of filters in traditional time or image based signal processing. Thus, comparing ours with existing work of graph CNNs, our paper provides a solid theoretical foundation for our proposed convolution step instead of an ad-hoc approach to convolution in CNNs for graph structured data.
>
> Further, our method avoids computing the spectrum of the graph Laplacian as in (Bruna et al. 2014), or approximating the spectrum using high degree Chebyshev polynomials of the graph Laplacian matrix (in Defferrard et al. 2016, it is suggested that one needs a 25th degree Chebyshev polynomial to provide a good approximation to the graph Laplacian spectrum) or using high degree Cayley polynomials of the graph Laplacian matrix (in Levie et al. 2017, 12th degree Cayley polynomials are needed). We also clarify that the GCN method in Kipf & Welling 2017 is a first order approximation of the Chebyshev polynomials approximation in Defferrard et al. 2016, which is very different from our method. Our method has a much lower computational complexity than the complexity of the methods proposed in Bruna et al. 2014, Defferrard et al. 2016, Levie et al. 2017, since our method only uses polynomials of the adjacency matrix with maximum degree 2 as shown in our experiments. Finally, the method that we propose exhibits better performance than existing methods.
>
> In the revised version, we have followed your suggestion and elaborated on the above two points as our main contributions and devoted more text to motivating our method.
>
> Bruna, J., Zaremba, W., Szlam, A., & LeCun, Y.  Spectral networks and locally connected networks on graphs, ICLR2013
> Defferrard, M., Bresson, X., & Vandergheynst, P. Convolutional neural networks on graphs with fast localized spectral filtering. In NIPS2016
> Graph Convolutional Neural Networks with Complex Rational Spectral Filters, submitted to ICLR18
> Kipf, T. N., & Welling, M. (2016). Semi-supervised classification with graph convolutional networks. ICLR2017

---

> > ### Author Response · Authors · 2017-12-30
> > **Reply to AnonReviewer 1: "Interesting improvement idea, clarity could be improved"-(Q2-Q3/Q11)**
> >
> > 2) "The main contribution seems to be the use of the "graph shift" operator from Sandryhaila and Moura (2013), which closely resembles the one from Shuman et al. (2013). It is actually not very well explained what is the main difference. "
> >
> > Reply: In Sandryhaila and Moura (2013), the graph shift operator is defined as the adjacency matrix, and the graph convolution operator is defined as multiplication by polynomials of the adjacency matrix, while in Shuman et al. (2013), the graph convolution operator is obtained by the eigendecomposition of the Laplacian matrix. There are in addition, significant differences between using the adjacency matrix or the graph Laplacian as “graph shifts.” The Laplacian matrix is a second order operator (like a second derivative) on the graph, and only applies to undirected graphs. In contrast, the shift operator is a first order operator (like a first-order derivative) on the graph and applies to arbitrary graphs (directed, undirected, or mixed).
> >
> > In addition, as indicated above, the previous work defined convolution in the spectral domain, rather than in the node domain, which may require either crude approximations of the spectrum or requires finding the spectrum, a costly operation, with computational complexity O(N^3), where N is the number of graph nodes. Thus, to avoid computing the spectrum of the Laplacian matrix, the spectrum is approximated and the convolution further approximated via different matrix polynomials, such as matrix Chebyshev polynomials in Defferrard et al. 2016 and matrix Cayley polynomials in Levie et al. 2017. To have reasonable approximate accuracy, very high degree of such special polynomials should be needed, which increases both the computation burden and taxes the numerical stability of the procedure. For example, in Defferrard et al. 2016 a Chebyshev polynomial of the Laplacian matrix with degree 25 is needed, and in Levie et al. 2017 a Cayley polynomial of the Laplacian matrix with degree 12 is needed to provide reasonable accuracy to the spectrum. In contrast, in our paper, we do not need computing the spectrum of the adjacency matrix and so no need to resort to these high degree polynomials. The graph filters we use only have a degree 2 and outperform the existing spectral based convolution methods in terms of classification accuracy, while not requiring costly polynomials of much larger degree.
> >
> > 3) "Equation (2) shows that the learnable filters g are operating on the k-th power of the normalized adjacency matrix A, so when K=1 this equals classical GCN from T. Kipf et al. By using K > 1 the method is able to leverage information at a farther distance from the reference node. "
> >
> > Reply: We agree with the reviewer that our method is able to leverage information at a farther distance on the graph than the GCN in Kipf & Welling 2017. However, ours is not a simple extension of GCN. In fact, extending GCN to the second order would not lead to our results.  In the separate comment (due to space limitation) with title “Differences between the proposed TAGCN and GCN in Kipf & Welling 2017”, we clarify the fundamental difference between our method and the GCN methodology if we extend the latter to a higher order. Thank you for your attention. We have added the corresponding discussion in Section 3 of the revised version.

---

> > > ### Author Response · Authors · 2017-12-30
> > > **Reply to AnonReviewer 1: "Interesting improvement idea, clarity could be improved"-(Q4-Q7/Q11)**
> > >
> > > 4) "Section 2.2 requires some polishing as I found hard to follow the main story the authors wanted to tell. The definition of the weight of a path seems disconnected from the main text, ins't A^k kind of a diffusion operator or random walk? This makes me wonder what would be the performance of GCN when the k-th power of the adjacency is used."
> > >
> > > Reply: We have polished section 2.2 as suggested. As A is the normalized adjacency matrix, A^k is indeed a weighted diffusion or random walk. In Section 2.2, we would like to understand the proposed convolution as a feature extraction operator in traditional CNN rather than as propagating labeled data on the graph. Taking this point of view helps us to profit from the design knowledge/experience from traditional CNN and apply it to grid structured data. Our definition of weight of a path and the following filter size (Section 2.2) for graph convolution make it possible to design a Graph CNN architecture similar to GoogLeNet (Szegedy et al., 2015), in which a set of filters with different sizes are used in each convolutional layer. In fact, we found that a combination of size 1 and size 2 filters gives the best performance in all three data sets studied, which is a polynomial with maximum order 2.
> > >
> > > As we explained above in the previous comment, graph convolution in our paper is not simply extending GCN to k-th order. To address the reviewer’s comment, nevertheless, we implement A^2 and compare its performance with ours. For the data sets Pubmed, Cora, and Citeseer, the classification accuracies are 79.1 (81.1), 81.7(82.5) and 70.8 (70.9), where the numbers in parentheses are the results obtained with our method. Our method still achieves a noticeable performance advantage over A^2 for the Pubmed and Cora data; in particular, we note the significant performance gain with the Pubmed database that has the largest number of nodes among these three data sets.
> > >
> > > 5) "I liked Section 3, however while it is true that all methods differ in the way they do the filtering, they also differ in the way the input graph is represented (use of the adjacency or not)."
> > >
> > > Reply: We agree with the reviewer and have incorporated this point of view in Section 3 in the revised version.
> > >
> > > 6) "Experiments are performed on the usual reference benchmarks for the task and show sensible improvements with respect to the state-of-the-art. "
> > >
> > > Reply: We would like to thank the reviewer for this comment. We also want to mention that, besides providing a solid foundation for our proposed graph convolution operation, our method also exhibits better performance due to the fact that no approximation is needed for the convolution operation. Our method outperforms all recently proposed methods on all three datasets. In addition, for the Pubmed dataset, which is much larger than the Citeseer and Cora data sets, we have a 2.1% improvement over GCN (Kipf & Welling 2017) and 6.7% improvement over ChebNet (Defferrard et al. 2016).  These performance results are averages obtained over 100 Monte Carlo runs.  As far as we know and as far as we can determine, our method exhibits the best performance on the Pubmed data not only when compared with all previously available publications, as well as when compared with all papers submitted to ICLR18, see papers below. Also, please note that, as explained by the authors, the last paper listed below fails with the Pubmed data set because of its storage complexity.
> > >
> > > Graph Partition Neural Networks for Semi-Supervised Classification, submitted to ICLR18
> > > Attention-based Graph Neural Network for Semi-supervised Learning, submitted to ICLR18
> > > Stochastic Training of Graph Convolutional Networks, submitted to ICLR18
> > > Graph Attention Networks, submitted to ICLR18
> > >
> > > 7) "TAGCN with K=2 has twice the number of parameters of GCN, which makes the comparison not entirely fair. Did the author experiment with a comparable architecture? Also, how about using A^2 in GCN or making two GCN and concatenate them in feature space to make the representational power comparable?  "
> > >
> > > Reply: We confirm that TAGCN with K=2 has twice the number of parameters in GCN. To compare implementations with the same number of parameters, we provide the performance of our method with half the number of filters in each convolution layer in the original submitted version (Table 4). These have the same number of parameters as GCN. Our method still has an obvious advantage in terms of classification accuracy. This proves that, even with a similar number of parameters or architecture, our method still exhibits superior performance than GCN. As explained in our response to previous comments, our method also still achieves a noticeable performance advantage over A^2 for the Pubmed and Cora data.
> > >
> > > In Appendix B of the original GCN paper, the authors have already extended the number of layers from 2 to 4, but their performance degrades when the number of layers is 4 as compared with the 2-layer case.

---

> > > > ### Author Response · Authors · 2017-12-30
> > > > **Reply to AnonReviewer 1: "Interesting improvement idea, clarity could be improved"-(Q8-Q11/Q11)**
> > > >
> > > > 8) "It is also known that these benchmarks, while being widely used, are small and result in high variance results. The authors should report statistics over multiple runs."
> > > >
> > > > Reply: Thank you for pointing out this. In the original submission, all the results are averaged performance over 100 Monte Carlo runs. We have added the statistics in our revised version.
> > > >
> > > > 9) "Given the systematic parameter search, with reference to the actual validation (or test?) set I am afraid there could be some overfitting. It is quite easy to probe the test set to get best performance on these benchmarks."
> > > >
> > > > Reply: We would like to clarify the reviewer on this point. As we explain in our experimental set up, we follow exactly the experimental settings in GCN. The data set is split into three parts: training, cross validation, and testing. We search the hyperparameters using cross validation on the validation set. And the performance results reported are evaluated on the test data set.
> > > >
> > > > 10) "As a minor remark, please make figures readable also in BW.”
> > > >
> > > > Reply: Thank you for your advice. We believe the reviewer refers to figure 2. The different colors represent filters at different locations. One can easily tell apart the different plots as they are in different figures. We have revised the description in the text to better reflect this and to make it easy to tell the differences.
> > > >
> > > > 11) Overall I found the paper interesting but also not very clear at pointing out the major contribution and the motivation behind it. At risk of being too reductionist: it looks as learning a set of filters on different coordinate systems given by the various powers of A. GCN looks at the nearest neighbors and the paper shows that using also the 2-ring improves performance.
> > > >
> > > > Reply: As in our response to the previous comments, this work is based on using graph filters designed from basic principles drawn from the graph signal processing, with no approximation of the graph convolution. GCN is based on approximations by Chebyshev polynomials. Further, extending GCN to 2-rings does not result in A^2. As we do not utilize approximations to the convolution operation, we obtain better classification accuracy when compared with any existing methods, either previously published or proposed in the current crop of papers submitted to ICLR18.

---

### Author Response · Authors · 2017-12-30
**Differences between the proposed TAGCN and GCN in Kipf & Welling 2017**

Our method is able to leverage information at a farther distance on the graph than the GCN of Kipf & Welling 2017. However, ours is not a simple generalization of GCN. Below, we clarify the fundamental difference between our method and the GCN methodology in Kipf & Welling 2017 if we extended the latter to a higher order:

Our first comment is that the graph convolution in GCN is defined as a first order Chebyshev polynomial of the graph Laplacian matrix, which is an approximation to the graph convolution defined in the spectrum domain in Bruna et al. 2014 (see eqn(4)-eqn(8) in Kipf & Welling 2017 for the derivation). In contrast, our graph convolution is rigorously defined as multiplication by polynomials of the graph adjacency matrix; this is not an approximation, rather, it simply is filtering with graph filters as defined and as being consistent with graph signal processing.

The approximate convolution by Chebyshev polynomials of the Laplacian matrix is defined as \sum_{k=0}^{K} \theta_k T_k(L) (eqn(5) in Kipf & Welling 2017), where T_k(L) is the matrix Chebyshev polynomials of degree k, and L is the graph Laplacian matrix. The matrix polynomial T_k(L) is recursively defined as T_k(L) = 2LT_{k-1}(L) – T_{k-2}(L) with T_0(L)=I and T_1(L) = L. This expression is K-localized, i.e., it depends only on nodes that are at maximum K steps away from the central node Kipf & Welling 2017. In Defferrard et al. 2016, 25th-order matrix Chebyshev polynomials are needed (K=25) for the semisupervised classification problem studied therein. Kipf & Welling 2017 adopted a first order matrix Chebyshev polynomial (K=1). By some further approximation, the convolution operator \sum_{k=0}^{K} \theta_k T_k(L) is approximated by \hat{A}, where \hat{A} is the normalized adjacency matrix of an undirected graph (see eqn(4)-eqn(8) in Kipf & Welling 2017).

Next, we show the difference between our work and the GCN method in Kipf & Welling 2017 when using 2nd order (K=2, 2 steps away from the central node) Chebyshev polynomials of Laplacian matrix following the above method. It has been shown that \sum_{k=0}^{1} \theta T_k(L) ≈\hat{A} in Kipf & Welling 2017, and T_2(L) =2L^2 -I by definition of Chebyshev polynomials. Then, the extension of GCN to the second order Chebyshev polynomials (two steps away from a central node) can be obtained from the original definition in Kipf & Welling 2017 (eqn (5)) as \sum_{k=0}^{2} \theta_k T_k(L)= \hat{A} + 2L^2 -I, which is obviously different from ours. Thus, it is evident that our method is not a simple extension of the GCN method in Kipf & Welling 2017. We apply graph convolution as proposed from basic principles in the graph signal processing, with no approximations involved, while both T. Kipf’s GCN and Bruna et al. 2014, Defferrard et al. 2016, Levie et al. 2017 are based on approximations of convolution defined in the spectrum domain. In our approach, the degree of freedom is the design of the graph filter – its degree and its coefficients. Ours is a principled approach and provides a generic methodology. The performance gains we obtain are the result of capturing the underlying graph structure with no approximation in the convolution operation.

Bruna, J., Zaremba, W., Szlam, A., & LeCun, Y. Spectral networks and locally connected networks on graphs, ICLR2013
Defferrard, M., Bresson, X., & Vandergheynst, P. Convolutional neural networks on graphs with fast localized spectral filtering. In NIPS2016
Graph Convolutional Neural Networks with Complex Rational Spectral Filters, submitted to ICLR18
Kipf, T. N., & Welling, M. Semi-supervised classification with graph convolutional networks. In ICLR2017

---

### Author Response · Authors · 2018-01-04
**new version uploaded**

We’ve uploaded a new version with the revised part in blue font. In the revised version, following the request of the reviewers, we make it clearer about the motivations and main contributions and move one subsection to the appendix. Besides, we thank the reviewer for pointing out one related paper and refer it in this new version.

---

### Decision · Program_Chairs · 2018-01-29
**ICLR 2018 Conference Acceptance Decision**

**Decision:**

Reject

**Comment:**

The authors provide an extension to GCNs of Kipf and Welling in order to incorporate information about higher order neighborhoods. The extension is well motivated (and  though I agree that it is not trivial modification of the K&W approach to the second order,  thanks to the authors for the clarification).  The improvements are relatively moderate.

Pros:
-- The approach is well motivated
-- The paper is clearly written
Cons:
-- The originality and impact (as well as motivation) are questioned by the reviewers